# Zoonotic Tuberculosis: A Neglected Disease in the Middle East and North Africa (MENA) Region

**DOI:** 10.3390/diseases11010039

**Published:** 2023-03-01

**Authors:** Dalal Kasir, Nour Osman, Aicha Awik, Imane El Ratel, Rayane Rafei, Imad Al Kassaa, Dima El Safadi, Rayane Salma, Khaled El Omari, Kevin J. Cummings, Issmat I. Kassem, Marwan Osman

**Affiliations:** 1Quality Control Center Laboratories at the Chamber of Commerce, Industry & Agriculture of Tripoli & North Lebanon, Tripoli 1300, Lebanon; 2Department of Epidemiology and Population Health, Faculty of Health Sciences, American University of Beirut, Beirut 1100, Lebanon; 3Laboratoire Microbiologie Santé et Environnement (LMSE), Doctoral School of Sciences and Technology, Faculty of Public Health, Lebanese University, Tripoli 1300, Lebanon; 4Fonterra Research and Development Center, Palmerston North 4410, New Zealand; 5Department of Clinical Sciences, Liverpool School of Tropical Medicine, Liverpool L3 5QA, UK; 6Department of Public and Ecosystem Health, College of Veterinary Medicine, Cornell University, Ithaca, NY 14853, USA; 7Center for Food Safety, Department of Food Science and Technology, University of Georgia, 1109 Experiment Street, Griffin, GA 30223, USA; 8Cornell Atkinson Center for Sustainability, Cornell University, Ithaca, NY 14853, USA

**Keywords:** *Mycobacterium bovis*, one health, epidemiology, antimicrobial resistance, MENA region

## Abstract

*Mycobacterium bovis* is the etiologic agent of bovine tuberculosis (BTB), a serious infectious disease in both humans and animals. BTB is a zoonotic disease primarily affecting cattle and occasionally humans infected through close contact with infected hosts or the consumption of unpasteurized dairy products. Zoonotic tuberculosis is strongly associated with poverty and poor hygiene, and low- and middle-income countries bear the brunt of the disease. BTB has been increasingly recognized as a growing public health threat in developing countries. However, the lack of effective surveillance programs in many of these countries poses a barrier to accurately determining the true burden of this disease. Additionally, the control of BTB is threatened by the emergence of drug-resistant strains that affect the effectiveness of current treatment regimens. Here, we analyzed current trends in the epidemiology of the disease as well as the antimicrobial susceptibility patterns of *M. bovis* in the Middle East and North Africa (MENA) region, a region that includes several developing countries. Following PRISMA guidelines, a total of 90 studies conducted in the MENA region were selected. Our findings revealed that the prevalence of BTB among humans and cattle varied significantly according to the population size and country in the MENA region. Most of the available studies were based on culture and/or PCR strategies and were published without including data on antimicrobial resistance and molecular typing. Our findings highlighted the paramount need for the use of appropriate diagnostic tools and the implementation of sustainable control measures, especially at the human/animal interface, in the MENA region.

## 1. Introduction

Bovine tuberculosis (BTB), caused by *Mycobacterium bovis*, is one of the world’s most neglected zoonotic diseases [1,2]. The prevalence of BTB follows a socioeconomic gradient by being concentrated in low- and middle-income countries (LMICs), mostly affecting poor, marginalized, and rural communities where people live in close contact with animals, and have limited access to sanitation, safe food and health care services [2]. Phylogenetically, *M. bovis* belongs to the *Mycobacterium tuberculosis* complex (MTBC), a cluster of genetically related *Mycobacterium* species that are associated with tuberculosis infections in a wide range of mammals [1]. Of all MTBCs, *M. bovis* is the most common cause of wildlife tuberculosis, with a morbidity risk reaching up to 15% of tuberculosis cases in humans [1,3]. This species has the widest host range, including domestic animals, livestock, wildlife, and humans [4]. The movement of animals is considered one of the main reasons for the spread of *M. bovis*, both within a country and across borders [2]. Indeed, the frequent movement (via trade) of cattle within and between countries and continents has facilitated the global spread of BTB [5,6]. Notably, the disease’s zoonotic property and its dynamic distribution has caused severe economic losses for dairy industries worldwide [7].

Although the dissemination of *M. bovis* has a heterogeneous profile, developed countries reported a significantly lower incidence of BTB infections compared to data from developing countries. The lack of effective policies to control BTB in many LMICs negatively affects the health of livestock, humans, and ecosystems and potentially increases the burden of this disease [6]. The transmission of BTB in humans is bipartite, (i) direct, through inhalation of the etiologic agent when in close contact with infected cattle or their carcasses, and (ii) indirect, associated with the consumption of unpasteurized dairy products or raw meat products from infected cattle [8,9,10,11]. The World Health Organization (WHO) developed the END-TB strategy to substantially reduce the annual number of tuberculosis deaths between 2016 and 2035 [12]. However, the COVID-19 pandemic has potentially adversely impacted progress in reducing the tuberculosis mortality rate, because the pandemic has disrupted access to essential resources for tuberculosis diagnosis and treatment [12]. To reinvigorate and facilitate control efforts, robust surveillance programs are needed, perhaps more than ever [13]. These programs are essential to better understand BTB transmission dynamics, bolster One Health policies, and predict future disease trends. Indeed, closing epidemiologic knowledge gaps is essential for a better understanding of the risk factors for transmission of BTB among vulnerable people, to support infection prevention and control and food safety policies, and to predict future disease trends in the Middle East and North Africa (MENA) region. Therefore, the main objective of this narrative review was to compile and discuss existing epidemiologic data on latent and active BTB in the MENA region, which includes many conflict-affected and economically challenged countries, with notable deficiencies in public health and national surveillance programs for the management and control of infectious diseases, including zoonotic diseases like BTB.

## 2. Methods

The burden of *M. bovis* infections in humans and animals is not well defined in the MENA region. Therefore, we searched PubMed, Science Direct, Scopus, and Google Scholar databases for epidemiologic studies on BTB published between 1990 and 2021. We used a combination of words that included “*Mycobacterium bovis*”, “Bovine”, “MENA countries (as described previously [14])”, “Epidemiology”, “Prevalence”, and “One Health”. After importation of the search results, two authors (D. Kasir and N. Osman) independently screened the citations for their relevance using the title and abstract and all qualified citations were retained for full-text assessment to confirm eligibility. Backward reference screening was performed for all articles. Data extraction was performed by the same authors through a format prepared on a Microsoft Excel workbook (Figure 1). Indexed original articles in English or French, of any epidemiologic design, sampling strategy, and type (case report, longitudinal, case-control, or cross-sectional) were included. All the studies that reported original information on the prevalence of BTB in MENA countries were eligible for inclusion in the review. We excluded narrative and systematic reviews. Given that this manuscript is a narrative review, no quality evaluation of the reviewed studies was performed.

## 3. Epidemiology of *Mycobacterium bovis* in the MENA Region

A total of 90 studies conducted in the MENA region were reviewed. Most studies have a cross-sectional design (87%; 78/90), followed by case reports (10%; 9/90) and longitudinal studies (3%; 3/90). We found that *M. bovis* has only been reported in humans and animals in eight MENA countries, including Algeria, Egypt, Iraq, Iran, Morocco, Sudan, Turkey, and Tunisia. In other MENA countries, Lebanon, Djibouti, Palestine, and Saudi Arabia, zoonotic tuberculosis has only been reported in humans. The epidemiologic trends of *M. bovis* infection varied across the MENA countries, likely influenced by the population size, characteristics of the targeted population, the geographical region, and the rigor of the adopted diagnostic tools and investigation methods. Additionally, the heterogeneity of BTB prevalence has been also associated with other factors such as Bacille Calmette-Guérin (BCG) vaccination status, the consumption of unpasteurized dairy products, and the efficiency of national surveillance programs and BTB control measures [15].

### 3.1. Mycobacterium bovis in Animals

Cases of BTB were noted from both pulmonary and extrapulmonary sites in animals (Table 1) and humans (Table 2). In animals, active BTB was usually reported in cattle and buffalo; however, uncommon cases were described among other types of animals. Specifically, *M. bovis* was reported in a cat and a mongoose in Turkey [16] and Egypt [17], respectively. A deer infected with BTB was also observed in Iran [18], while *M. bovis* was detected in camels and pigs in Egypt [19,20].

Several risk factors for BTB appear to play an essential role in the spread of *M. bovis* among animals in the MENA region. Age, gender, animal body condition, immune suppression, crowding, cross-species transmission, grazing practices, feeding system, environment or weather, and physiological and pathological variations are potential factors contributing to the dissemination of zoonotic *M. bovis*. Female animals are at a greater risk of BTB than males due to lactation, gestation, and parturition [21,22]. Cross-species transmission between goats and cattle and between buffalo and cattle was associated with sharing of drinking and grazing locations in Algeria [23] and Iran, respectively [24,25]. Furthermore, uncontrolled animal migrations and trade within and across countries were noted as key drivers for BTB transmission [26]. People working closely with livestock, particularly dairy cattle (e.g., farmers, veterinarians, slaughterhouse workers) or with wildlife were more susceptible to *M. bovis* infections [27].

The World Organization for Animal Health (WOAH) has categorized the tuberculin skin test (TST) as a primary screening test for tuberculosis in cattle [28]. TST is the most frequently used test for the diagnosis of BTB in cattle. Typically, TST’s discriminatory power could be improved by combining it with the interferon-gamma release assay (IGRA) which improves both sensitivity and specificity [29]. *M. bovis* ELISA tests are also available, allowing the detection of antibodies against zoonotic tuberculosis in cattle serum and plasma samples [30]. Although the ELISA assay is not yet recognized as a standard test for tuberculosis in cattle, it has been approved by the WOAH as being complementary to the TST in cattle. It should be noted that when using these diagnostic approaches, it is difficult to distinguish between vaccinated and infected animals and latent and active infections [31]. However, based on these assays, the prevalence of BTB among cattle varied significantly according to the population size and country in the MENA region. In large studies, the prevalence was relatively low, ranging between 0.1 [32] and 16.4% [33] in Egypt, 4.4 [34] and 24.2% [35] in Iraq, 3.5% [36] in Algeria, and 1.4% [37] in Turkey. In contrast, a higher prevalence was reported in studies with small population sizes, ranging from 22.2% [38] to 82.6% [39] in Egypt, 75% [40] in Iraq, and 48% [41] in Tunisia (Table 1). The trends in the prevalence of BTB also changed over time. In Egypt, Iran, Iraq, Morocco and Sudan, the prevalence of infection among cattle varied between 0.2% [42] and 4.3% [43], 8.5% [44] and 26.3% [45], 1.3% [46] and 10.2% [47], 1.7% [48] and 51.3% [49] and 0.2% [50] and 20.8% [51] over the last two decades, respectively.

Regarding *M. bovis* in milk samples collected in the MENA region, most studies reported a relatively low prevalence, ranging from 0.004% [33] to 10.2% [52]. Notably, Iraq and Tunisia led the list of *M. bovis* prevalence in milk samples (Table 1). Using the ELISA assay, a higher infection risk (20.2%) among lactating cows was found in rural areas of Waist and Dhi-Qar provinces, Iraq [53]. Despite the challenges in detection of *M. bovis* in milk samples, available data from the MENA region confirmed that this matrix represents an important source of zoonotic tuberculosis, because milk (1) is still commonly consumed raw, without pasteurization, in many rural regions and (2) is widely used in the manufacturing of popular dairy products such as cheese and yogurt [54]. Taken together, available data underlined the existence of animal and food sources as well as zoonotic risks that escaped common tuberculosis control measures in many MENA countries. Therefore, there is a strong need to increase awareness on food safety and hygiene and strengthen active surveillance programs in food animals and their products [55]. To prevent the further dissemination of BTB infection, effective approaches must be adopted, including early identification, adequate therapy, and contact tracing [56]. Currently, BTB control mostly relies on slaughter policy, postmortem inspection, and slaughterhouse surveillance [57], which do not even address preharvest risks.

**Table 1 diseases-11-00039-t001:** Burden of *Mycobacterium bovis* in animals in the MENA region.

Country	Study Period	Study Design	Population (N)	Tuberculosis (TST, IGRA, ELISA) ^†^	Samples (N) for Active Tuberculosis Testing	Health Status	BTB Identification (Culture, PCR)	Prevalence ^¶^ of BTB (%)	Typing Method	References
Algeria	2007	Cross-sectional	Cattle (7250)		Tissue (260)	Slaughtered	Culture	88 (1.2%)	Spoligotyping; MIRU-VNTR	[31]
	2017	Cross-sectional	Cattle (3848)		Tissue (3848)	Slaughtered	Culture; PCR	59 (1.5%)	Spoligotyping; MIRU-VNTR	[58]
	2017–2018	Cross-sectional	Cattle (928)		Tissue (928)	Slaughtered	Culture; PCR	13 (1.4%)	WGS	[21]
	2017–2019	Cross-sectional	Cattle (3546)		Tissue (3546)	Slaughtered	Culture; PCR	174 (4.9%)	Spoligotyping	[23]
	2018–2019	Cross-sectional	Cattle (516)	18 (3.5%)		Live		ND		[36]
Egypt	2008–2010	Cross-sectional	Cattle (3255) Buffalo (2950)	Cattle: 105 (3.2%) Buffalo: 85 (2.9%)	Tissue (190) Milk (520) Blood (190)	Slaughtered	Culture; PCR	16 (0.2%)		[59]
	2008–2010	Cross-sectional	Cattle (1180)	29 (2.5%)	Tissue (29)	Slaughtered	Culture; PCR	20 (1.7%)		[60]
	2010–2011	Cross-sectional	Cattle (3347)	32 (1%)	Tissue (32)	Live and slaughtered	Culture; PCR	21 (0.6%)		[61]
	2013	Cross-sectional	Cattle		Milk (100)	Healthy	Culture; PCR	1 (1%)		[62]
	2014–2015	Cross-sectional	Cattle (2935)	63 (2.2%)	Tissue (56)	Slaughtered	Culture	39 (1.3%)		[63]
	2008 *	Cross-sectional	Camels (704)	9 (1.27%)	Tissue (29)	Slaughtered	Culture	5 (0.7%)		[64]
	2009 *	Cross-sectional	Cattle (46)	38 (82.6%)	Milk (23)	Sick	Culture	1 (2.1%)		[39]
	2014 *	Cross-sectional	Cattle (422) Buffalo (480)	Cattle: 9 (2.1%) Buffalo: 27 (5.62%)	Tissue (36)	Slaughtered	Culture	25 (2.8%)	IS*6110* RFLP	[65]
	2015 *	Cross-sectional	Cows (420)	8 (1.9%)	Milk (8)	Healthy	Culture; PCR	1 (0.2%)		[42]
	2018 *	Cross-sectional	Sheep (18)	4 (22.2%)	Tissue (18)	Slaughtered	Culture; PCR	15 (83.3%)		[38]
	2009–2013	Longitudinal	Cows and buffalos (1,186,772)	1225 (0.1%)	Blood (14) Tissue (34)	Live and slaughtered	Culture; PCR	29 (0.002%)		[32]
	2016–2019	Cross-sectional	Cattle (2200) Buffalo (1500)		Tissue		Culture; PCR	Cattle 36 (1.6%) Buffalo 18 (1.2%)		[66]
	2018	Cross-sectional	Cattle (2650)	63 (2.4%)	Tissue (63)	Healthy	Culture	47 (1.8%)		[67]
	2018–2019	Cross-sectional	Cattle (569) Buffalo (181)		Tissue (30)	Slaughtered	Culture; PCR	9 (1.2%)		[68]
	2011–2016	Cross-sectional	Cattle (1570) Buffalo (530)	74 (3.5%)	Tissue (74)	Slaughtered	PCR	61 (2.9%)		[69]
	2017	Cross-sectional	Cattle (2710)	215 (7.9%)	Milk (245)	Live	Culture; PCR	68 (2.5%)		[70]
	2014 *	Cross-sectional	Cattle (300)	53 (17.6%)	Blood (65)	Live and slaughtered	Culture; PCR	13 (4.3%)		[43]
	2011	Case report	Mongoose (1)		Tissue (1)	Slaughtered	Culture; PCR	1		[17]
	2015–2017	Longitudinal	Camels (10,903)	184 (1.7%)	Tissue (184)	Live and slaughtered	Culture; PCR	112 (1.0%)		[19]
	2018–2019	Cross-sectional	Cattle (1464)		Milk (1285); Lymph nodes (179)	Live and slaughtered	Culture; PCR	127 (8.6%)		[71]
	2011–2016	Cross-sectional	Cattle and Buffalo (2100)	81 (3.8%)	Tissue	Live	Culture; PCR	61 (2.9%)	MIRU-VNTR	[26]
	2016 *	Cross-sectional	Cattle and Buffalo (6000)	79 (1.3%)	Tissue	Live and slaughtered	Culture; PCR	23 (0.4%)		[72]
	2004–2005	Cross-sectional	Pigs (745)		Tissue	Slaughtered	Culture; PCR	12 (1.6%)		[20]
	2019 *	Cross-sectional	Cattle (2600)	47 (1.8%)	Tissue	Healthy	Culture; PCR	40 (1.5%)		[73]
	2006–2008	Cross-sectional	Cattle (3000)	108 (3.6%)	Tissue	Slaughtered	PCR	90 (3%)		[74]
	2013 *	Cross-sectional	Cattle (3474)	78 (2.2%)		Slaughtered		ND		[75]
	2013–2015	Cross-sectional	Cattle (7064)	242 (3.4%)	Tissue	Slaughtered	Culture; PCR	31 (0.4%)	MIRU-VNTR; WGS	[76]
	2020 *	Cross-sectional	Cattle (50)	50 (100%)	Tissue	TST-positive	Culture; PCR	45 (90%)		[77]
	2017	Cross-sectional	Cattle (2710)	444 (16.4%)	Blood and milk (444)	TST-positive	Culture; PCR	Blood: 44 (1.6%); Milk: 12 (0.004%)		[33]
Iran	2003–2005	Cross-sectional	Buffalo (140)		Tissue (140)	Slaughtered	Culture	0	RFLP	[45]
	2003–2006	Cross-sectional	Cattle (213)		Tissue (213)	Slaughtered	Culture; PCR	56 (26.3%)	RFLP; MIRU-VNTR; Spoligotyping	
	1996–2006	Cross-sectional	Cattle (488); Buffalo (140)		Tissue	Slaughtered	Culture; PCR	Cattle: 67 (13.7%); Buffalo: 132 (28.1%)	RFLP; RD-PCR; MIRU-VNTR	[24,25]
	2016 *	Case report	Deer (1)		Tissue	Dead	PCR	1	IS*6110* RFLP	[18]
	2016	Cross-sectional	Cattle (1700)		Tissue	Healthy	PCR	44 (8.5%)		[44]
Iraq	2009 *	Cross-sectional	Cattle		Milk (68)		Culture; PCR	7 (10.2%)		[47]
	2016 *	Cross-sectional	Cattle (300)		Tissue	Slaughtered	Culture	4 (1.3%)		[46]
	2015–2016	Cross-sectional	Cows (119)	24 (20.2%)	Blood and milk	Live		42 (35.2%)		[53]
	2019	Cross-sectional	Cattle (106); Buffalo (90)	Cattle (12.2%); Buffalo (4.4%)		Live		ND		[34]
	2010	Cross-sectional	Cattle		Milk (102)	Healthy	Culture; PCR	10 (9.8%)		[52]
	2016	Cross-sectional	Cattle (186)	32 (17.2%)		Live		ND		[78]
	2014 *	Cross-sectional	Cattle (28)	21 (75%)		Slaughtered		ND		[40]
	2012 *	Cross-sectional	Cows (850)	206 (24.2%)	Serum (260), Milk (45), swab nasal (45), tissue samples (98), from cattle	Live and slaughtered	Culture	100 (11.8%)		[35]
	2016 *	Cross-sectional	Cattle (21)	4 (19%)		Live		ND		[79]
Morocco	2014–2015	Cross-sectional	Cattle (8658)		Tissue	Slaughtered	Culture; PCR	144 (1.7%)	Spoligotyping	[48]
	2018 *	Cross-sectional	Cattle (1087)	222 (20.4%)		Live		ND		[80]
	2000–2001	Cross-sectional	Cattle (78)		Tissue	Slaughtered	Culture	40 (51.3%)		[49]
	1990	Cross-sectional	Cattle (246)	114 (46.3%)	Blood and Tissue	Live and slaughtered	Culture	73 (29.7%)		[81]
Sudan	2007–2009	Cross-sectional	Cattle (6680)		Tissue	Slaughtered	Culture; PCR	12 (0.2%)		[50]
	2002 *	Cross-sectional	Cattle (120)		Lymph nodes and tissue	Slaughtered	Culture; PCR	25 (20.8%)	IS*6110* RFLP	[51]
Turkey	2019 *	Case report	Cat (1)		Tissue	Slaughtered	Culture; PCR	1		[16]
	2008	Cross-sectional	Cattle (145)		Milk (145)	Live	Culture; PCR	1 (0.7%)	Spoligotyping	[82]
	2011–2012	Cross-sectional	Cattle (5018)		Tissue (95)	Slaughtered	Culture	32 (0.6%)	Spoligotyping; MIRU-VNTR	[83]
	2005	Cross-sectional	Cattle (210)	3 (1.4%)	Nasal (198); Milk (146)	Live	PCR	3(1.42%)		[37]
	2017–2018	Cross-sectional	Cattle (ND)		Lymph nodes and tissue	Slaughtered	Culture	38 (ND)	EIRC-PCR; RAPD-PCR; OUT-PCR; Spoligotyping	[84]
Tunisia	2005–2006	Cross-sectional	Cattle (102)		Milk (306)	TST-positive	Culture; PCR	5 (4.9%)	IS*6110* RFLP; Spoligotyping; MIRU-VNTR	[85]
	2014–2015	Cross-sectional	Cattle (149)		Tissue (149)	Slaughtered	Culture	96 (64.4%)	IS*6110* RFLP; Spoligotyping; MIRU-VNTR	[86]
	2010–2011	Cross-sectional	Cattle (100)	48 (48%)	Tissue (100)	Slaughtered	Culture; PCR	27 (27%)	Spoligotyping; MIRU-VNTR	[41]

* Date of publication; ^†^ Based on interferon gamma release assay, ELISA, or tuberculin skin test (TST). If different methods were used, we adopted the results of the TST; ND, Not Determined; MIRU-VNTR, Mycobacterial Interspersed Repetitive Units—Variable Number of Tandem Repeats; MLVA, Multiple Locus Variable Number of Tandem Repeats Analysis; ETR, Exact Tandem Repeats; RFLP-PCR, Restriction Fragment Length Polymorphism-PCR; WGS: Whole Genome Sequencing. ^¶^ Prevalence = n of *M. bovis* infected cases/n of total population.

### 3.2. Mycobacterium bovis in Humans

In humans, *Mycobacterium tuberculosis* is the primary causative agent of tuberculosis, followed by other MTBC species, including *M. bovis*. Nationwide estimations in the MENA countries, when available, revealed a relatively low prevalence of *M. bovis* among tuberculosis patients in some countries (Table 2). The prevalence of *M. tuberculosis* and *M. bovis* in Turkey was 94.1% and 4.3%, respectively [87]. Similarly, a study showed that only one tuberculosis case was due to *M. bovis* out of 67 extrapulmonary [88] and 45 pulmonary [89] tuberculosis cases in Egypt. In Lebanon, a nationwide surveillance study on tuberculosis showed that 3.4% (12/348) of patients were infected with *M. bovis*, while the remaining cases had human-associated tuberculosis strains (i.e., *M. tuberculosis* or *Mycobacterium africanum*) [90]. In contrast, BTB appears to have rapidly increased in comparison to other forms in Tunisia in recent years. Specifically, the estimated prevalence increased from 2.2% in 2009 [91] to 92.4% in 2013 [92]. Additionally, when focusing on at-risk groups such as farmers or slaughterhouse employees, available data showed high proportions of zoonotic tuberculosis, ranging from 8% in Iraq [35] and 5.36% in Egypt [71], to 3.3% in Lebanon [93].

The paucity of data and deficiencies in rigorous monitoring along with inappropriate control measures might cause the disease to spread more within the MENA region and beyond. Several factors might promote the spread of BTB among humans in the MENA region. For example, inappropriate hand washing or disinfection following cow handling appears to be a major risk factor for *M. bovis* infections among dairy farm workers [42]. The consumption of contaminated raw or unpasteurized milk also plays a crucial role in the transmission of BTB and has been significantly associated with the elevated risk of *M. bovis* infections in dairy workers [20,42,94]. The latter might also affect other human and animal populations. For instance, in Turkey and Lebanon, raw milk is widely available, which increases the risk of becoming infected from contaminated milk [84]. Furthermore, close quarters and proximity to animals, inadequate ventilation, and cow crowding were significant contributors to an increase in the risk of BTB [42,68]. These conditions are relevant in rural regions and in refugee camps in several Middle Eastern countries (e.g., Lebanon, Jordan, Turkey) and some geographical locations (e.g., the Nile Delta and Valley in Egypt) [63,95,96,97].

**Table 2 diseases-11-00039-t002:** Burden of *Mycobacterium bovis* among humans in the MENA region.

Country	Study Period	Study Design	Population (N)	Samples (N) for Active Tuberculosis Testing	Health Status	Identification Method	Prevalence ^¶^ of BTB (%)	Typing Method	References
Algeria	2015–2018	Cross-sectional	ND (98)	Sputum (98)	Pulmonary TB		4 (4.3%)	WGS	[98]
	2017–2019	Cross-sectional	ND (1952)	Sputum (51), Bronchial aspiration fluids (7); Gastric aspirations (25); Extra-pulmonary specimens (32)	TB patients	Culture; PCR	7 (0.3%)	Spoligotyping; PhyloSNP	[23]
Egypt	1998–2000	Cross-sectional	ND (67)	Cerebrospinal fluid (67)	Meningitis patients		1 (1.5%)	IS*6110* RFLP; Spoligotyping	[88]
	2010–2011	Cross-sectional	ND (42)	Sputum (42)	TB patients	Culture; PCR	0		[61]
	2013	Cross-sectional	Dairy workers	Hand swab (50)	Healthy	Culture; PCR	0		[62]
	2007 *	Cross-sectional	ND (45)	Sputum (45)	Pulmonary TB	PCR	1 (2.2%)	IS*6110* RFLP; Spoligotyping	[89]
	2009 *	Cross-sectional	Farm workers (15)	Sputum (15)	Healthy	Culture			[39]
	2015 *	Cross-sectional	Farm workers (25)	Sputum (25)	Healthy	Culture; PCR	1 (4%)		[42]
	2018 *	Cross-sectional	Farm workers (10)	Blood (10)	Healthy	Culture; PCR			[38]
	2015–2017	Longitudinal	Humans in contact with camels (48)	Sputum (48); Serum (48)	Healthy	Culture; PCR	0		[19]
	2018–2019	Cross-sectional	Farm workers (149)	Sputum (149)	Healthy	Culture; PCR	8 (5.3%)		[71]
	2016 *	Cross-sectional	ND (10)	Sputum (3)	Diagnosed human	Culture; PCR	0		[72]
	2020 *	Cross-sectional	ND	Sputum (10)	Tuberculin test positive	Culture; PCR	90%	Sequencing (*Mpb70* genes)	[77]
Iran	2009	Cross-sectional	ND	Mycobacteriology bank in MRC (60)		Culture	1 (1.7%)	MIRU-VNTR Spoligotyping	[99]
	2004–2005	Cross-sectional	ND (165)	Positives isolates (156)	TB patients		15 (9.7%)	IS6110-RFLP MIRU-VNTR ETR-VNTR	[100]
	1995–2004	Cross-sectional	ND (30)	Serum (30)	Patients with disseminated BCG disease		17 (56.6%)		[101]
	2016	Case report	ND (1)	Tissue	Brain tuberculoma	PCR	1		[102]
Iraq	2016 *	Cross-sectional	ND (186)	Sputum (186)	Healthy	Culture	2 (1.1%)		[46]
	2012 *	Cross-sectional	Farm workers and veterinary doctors (25)	Sputum (25); Serum (25)	Healthy	Culture	2 (8%)		[35]
Lebanon	2004–2005	Cross-sectional	Workers and veterinary doctors (60)	Sputum (60)	Pulmonary TB	Culture; PCR	2 (3.3%)	Spoligotyping	[93]
	2015–2017	Cross-sectional	ND (13)	Clinical samples (13)	Suspected TB patients	Culture	2 (15.4%)	IS*6110* insertion; Spoligotyping; MIRU-VNTR; WGS	[103]
	2016–2017	Cross-sectional	ND (1104)	Clinical samples (1104)	TB patients	Culture; PCR	12 (1.1%)	Spoligotyping; MIRU-VNTR; Deeplex-TB	[90]
Morocco	2000–2001	Cross-sectional	ND (200)	Sputum (200)	Suspected TB patients	Culture	18 (17.8%)		[49]
	2011 *	Case report	ND (1)	Gastric specimen	Patient with erythema nodosum	Culture	1		[104]
Palestine	2005–2010	Cross-sectional	ND (53)	Sputum (53); Smears (31)	TB patients	Culture	2 (3.7%)	Spoligotyping MIRU-VNTR	[105]
Djibouti	1999	Cross-sectional	ND (153)	Lymph nodes (196)	Patients with adenopathy	Culture	1 (0.7%)		[106]
	1997–2011	Cross-sectional	ND (411)	Sputum (411)	Suspected TB patients	Culture	1 (0.2%)	Spoligotyping; VNTR-MLVA; WGS	[107]
Saudi-Arabia	2002–2005	Cross-sectional	ND (1505)	Clinical isolates (1505)	Healthy	Culture	13 (0.9%)	Spoligotyping; MIRU-VNTR	[108]
	2014–2016	Cross-sectional	ND (2092)	Extrapulmonary clinical isolates (1003); Pulmonary clinical isolates (1089)	TB patients	Culture	Extrapulmonary: 119 (11.8%); Pulmonary: 32 (2.9%)	MIRU-VNTR	[109]
Sudan	1998–1999	Cross-sectional	ND (105)	Sputum (105)	TB patients	PCR	1 (0.9%)	Spoligotyping	[110]
Turkey	2007–2010	Cross-sectional	ND (188)	Clinical samples (188)	TB patients	Culture; PCR	8 (4.3%)		[87]
	2011–2012	Cross-sectional	ND (10)	Sputum (10)	TB patients	Culture	5 (50%)	Spoligotyping; MIRU-VNTR	[83]
	2015 *	Case report	ND (1)	Tissue sample	NEMO-deficient patient	Culture; PCR	1	GenoType MTBC; Spoligotyping	[111]
	2007–2010	Cross-sectional	ND (2436)	Clinical samples (188)	TB patients	PCR	8 (0.3%)	GenoType MTBC	[87]
	2016	Case report	Slaughterhouse worker (1)	Clinical sample	Skin lesion	Culture	1	GenoType MTBC	[112]
	1996 *	Case report	Slaughterhouse worker (1)	Clinical sample	Flexor Tenosynovitis	Culture	1		[113]
	2007 *	Cross-sectional	ND (60)	Sputum (60)	TB patients	PCR	8 (13.3%)		[114]
	2004–2014	Cross-sectional	ND (220)	Clinical samples (220)	TB patients	Culture	3 (1.4%)	Genotyping MTBC	[115]
	2009–2014	Cross-sectional	ND (482)	Clinical samples (482)	Pulmonary and extrapulmonary TB patients	Culture	13 (2.7%)	Spoligotyping	[94]
Tunisia	2014–2018	Case report	ND (4)	Tissue (4)	Spondylodiscitis patients	Culture; PCR	4		[116]
	2009–2013	Cross-sectional	ND (181)	Tissues (181)	Patients with adenopathy	Culture	4 (2.2%)		[91]
	2013	Cross-sectional	ND (174)	Lymph node (174)	Patients with adenopathy	Culture; PCR	60 (34.4%)		[117]
	2013–2015	Cross-sectional	ND (170)	Lymph nodes biopsy (144); Pus and abscess (10); Cerebrospinal fluid (8); Pleural fluid (1); Tissue (5); Bone scarping (2)	TB patients	Culture; PCR	157 (92.4%)		[92]

* Date of publication; If both were used, we adopted the results of the interferon gamma release assay; ND, Not Determined; MIRU-VNTR, Mycobacterial Interspersed Repetitive Units—Variable Number of Tandem Repeats; MLVA, Multiple Locus Variable Number of Tandem Repeats Analysis; ETR, Exact Tandem Repeats; RFLP-PCR, Restriction Fragment Length Polymorphism-PCR; WGS: Whole Genome Sequencing. ^¶^ Prevalence = n of *M. bovis* infected cases/n of total population.

### 3.3. Laboratory Methods for the Diagnosis and Typing of Mycobacterium bovis Adopted in the MENA Region

Although the reported detection methods in the studies from the MENA countries varied, active tuberculosis infections are still confirmed by mycobacterial culture which is considered the main approach, even for BTB. However, the adoption of molecular assays might be advantageous. For example, comparing molecular assays with microbiological culture revealed that the detection level of PCR-based assays was slightly greater than the conventional culture approach [118]. Moreover, molecular methods provide faster detection and identify the isolates at the species level. To confirm the identification of *M. bovis*, the detection of polymorphism in *pncA* or *oxyR* genes represents a valuable approach [119,120]. Recently, two PCR-based methods, VetMAX^TM^ and GeneXpert^®^, were developed for *M. bovis* identification [1,121].

Traditionally, the molecular epidemiology of *M. bovis* is studied by DNA fingerprinting methods such as IS*6110* RFLP (Restriction Fragment Length Polymorphism) [122]. Despite the method’s potential to identify outbreaks in hospitals and communities, its low discriminatory power for strains with low number of IS*6110* copies imposes the need for other complementary tools, such as spoligotyping and Mycobacterial Interspersed Repetitive Units/Variable Number Tandem Repeat (MIRU/VNTR) [123,124]. Furthermore, next-generation genome sequencing is receiving significant attention for *M. bovis* diagnosis because it provides a higher discriminatory power, facilitating the investigation of MTBC molecular epidemiology and genetic diversity with greater resolution [125]. However, when sequencing is unavailable (due to limited resources in LMICs), the combination of MIRU-VNTR with spoligotyping is more suitable for tracking infections and detecting risk factors than either technique alone [99]. Regardless, the application of molecular techniques in the genotyping of *M. bovis* facilitates infection control and tracking processes. An obvious example of the latter is revealing the effect of the animal movement on the appearance of *M. bovis* in African countries, which was mainly due to cattle delivered from Europe. This cross-border link was detected by using the spoligotyping approach, with SB0120 and SB0121 spoligotypes being the most abundant of *M. bovis* [109]. Spoligotype SB0120 is the most common circulating type worldwide while SB0121 mainly exists in Europe [126]. This geographical spillover was also observed in the MENA countries, especially Tunisia, Algeria, Morocco [34,98,116] and Iran [24].

### 3.4. Antimicrobial Resistance among M. bovis Isolates in the MENA Region

A major factor that might complicate the control of *M. bovis* in the MENA region is the drug resistant properties of this zoonotic agent. *M. bovis* has a natural resistance to pyrazinamide, an essential drug for standard short-course anti-tuberculosis therapy in humans. Unfortunately, phenotypic susceptibility to pyrazinamide is often not tested in the MENA region. Since the currently adopted diagnostic tools do not usually differentiate *M. bovis* from other MTBC species in MENA countries, BTB patients receive inadequate treatment, risking poorer outcomes and enhancing the selection of drug-resistant strains. Additionally, alarming data on drug resistance have been reported recently in the MENA region. Antimicrobial resistance genes were found in isolates retrieved from both infected humans and animals [95,127,128]. *M. bovis* strains were most commonly resistant to rifampicin and isoniazid in several reports from the MENA region [71,110,111]. A rifampicin (RIF)-resistant *M. bovis* strain was first reported in a Turkish patient in 2015, an 8-month-old male infant with nuclear factor-kB essential modulator (NEMO) deficiency [111]. Moreover, an Egyptian study reported the spread of multidrug-resistant *M. bovis* strains among buffaloes [68]. In Sudan, 4% of *M. bovis* isolates possessed resistance to both rifampicin and isoniazid due to genetic mutations [110], while, in Palestine, mutated *rpoB* and *katG* genes were identified in clinical samples from three unrelated individuals who did not respond to the first line of antituberculosis drug therapy [105].

## 4. Concluding Remarks

To our knowledge, this is the first review regarding the epidemiology of *M. bovis* in the MENA region. Despite the limited number of studies dealing with the epidemiology of *M. bovis* in most MENA countries, the currently available data shows that BTB is not negligible. The circulation of *M. bovis* in the community, even at a comparatively low prevalence, emphasizes the global and regional calls for appropriate diagnosis and control measures at the animal/human interface. In some MENA countries, the literature suggests that the prevailing conditions might be conducive for the spread of *M. bovis* in humans and other animals. This is facilitated by several factors, including the lack of information and deficient diagnosis and monitoring systems [42,49]. Therefore, it is obvious that the underdiagnosis of *M. bovis* in the MENA region emphasizes the need to review the current policies and guidelines adopted by public health stakeholders. Particularly, a better detection of zoonotic tuberculosis cases is required via enhancing laboratory capacity, ensuring access to fast and reliable diagnostic tools, raising awareness and expertise of stakeholders, improving food safety, strengthening surveillance (especially in animals), and addressing research gaps. Finally, the implementation of a One Health approach is crucial to control the spread of this disease in the MENA countries and beyond.

## Figures and Tables

**Figure 1 diseases-11-00039-f001:**
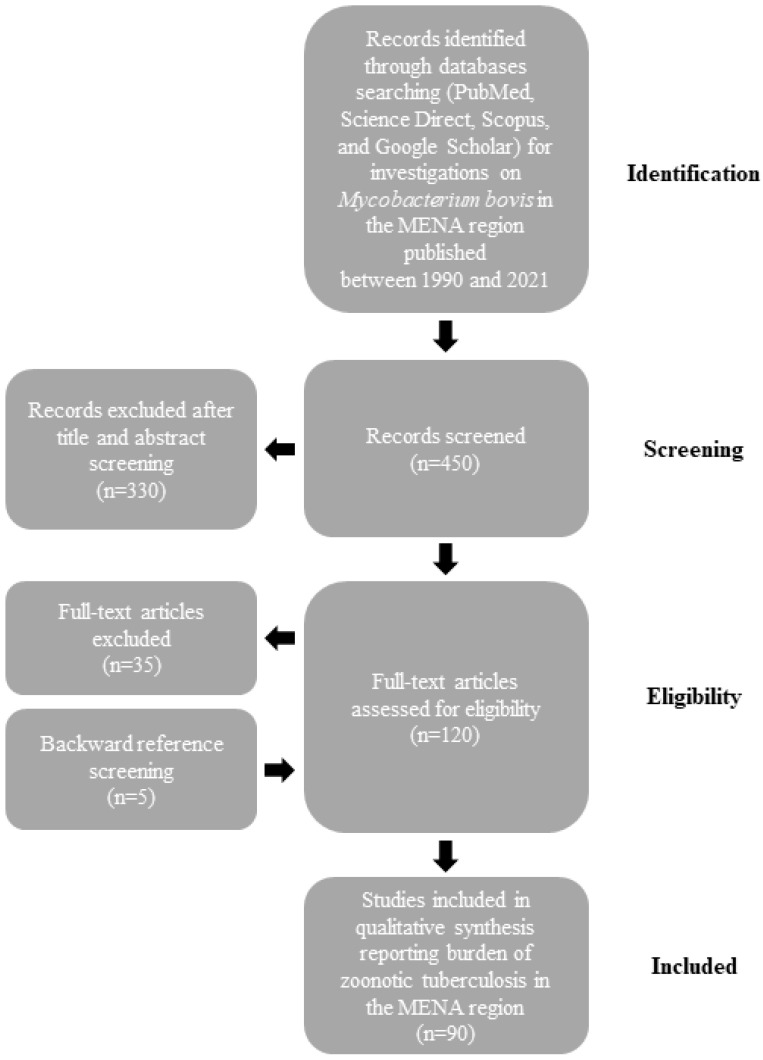
Flow diagram describing paper selection and inclusion/exclusion process for the review according to PRISMA guidelines.

## Data Availability

Not applicable.

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
