# Peer review of "Zoonotic Tuberculosis: A Neglected Disease in the Middle East and North Africa (MENA) Region"

_diseases, 2023, doi:10.3390/diseases11010039_

Round 1

Reviewer 1 Report

Dear authors,

thank you for the present manuscript. As the title suggest, it focus on a very important topic that is too often neglected. Given the relevance of the topic, I would suggest resubmitting the manuscript after an extensive and profound revision.

For the next time, providing the manuscript as text file would facilitate the revision process.

Abstract

The abstract focuses on the background and lack methods and results of the study. Please, consider re-writing it to give readers a summary of what’s inside the manuscript instead of an introduction to the topic.

Introduction

Lines 39-40: “and it is the most common form of wildlife tuberculosis, with a morbidity risk reaching up to 15% of tuberculosis cases in humans” How do the two sentences relate to each other? What does wildlife TB have to do with human morbidity? Please, consider separating the sentences.

Lines 65-66: “To reinvigorate and facilitate control efforts, robust surveillance programs are needed, perhaps more than ever.” Citation required or, if it is authors’ opinion, argument it in the discussion section.

Please, highlight the knowledge gap that makes your manuscript valuable for the scientific community working on this topic.

Methods

This section lacks details that would allow readers to reproduce the results, i.e., the complete list of research queries. Please, provide the query string used for each search engine with research terms and logic operators.

Results

Line 100: “unexpected cases”. Are those cases unexpected because they fell out of the M. bovis host-range? Or is it authors’ opinion? Please, argument or remove it for scientific soundness.

Please, provide a summary of the type of research work that were considered for this review. How many cross-sectional studies? Longitudinal ones? And case-control studies?

No risk measures are provided for risk factors. No confidence intervals are provided for any of the risk factors or for the prevalence proportions listed.

The text is structured like in a systematic review with materials and methods, results, conclusion, but it is not. Please consider restructuring the whole manuscript in a more discursive way, avoiding such structure and grouping results by means of the topic (transmission, zoonotic potential, cross-border spreading, diagnostics etc.) instead. Also, it would be useful to provide risk measures along with confidence intervals, or to suggest the degree of confidence based on the type of study from which the information was obtained.

Lines 132-134: Please, move the references at the end of the sentence like “0.2-4.3% [41-42], 8.5-26.3% [43-44], 1.3-10.2% [45-46], 1.7-51.3% [47-48], and 0.2-20.8% [49-50]” to improve readability. Also, provide confidence intervals for each estimate, otherwise they are not comparable.

Line 135: “Overall, we noticed that the prevalence has decreased in these countries over time (Table 1).” How did you come to this conclusion? Did you divide results by animal species? Did you calculate trend statistics? If so, please provide details of the method and statistical results. If not, describe clearly where you identified the decreasing trend.

Lines 163-164: “Prevalence = n of M. bovis infected cases/ n of total population.” The prevalence here is difficult to understand since the population which authors refer to varied, depending on the study, from 1 to 7250, and also include non-numerical values like “dairy workers”. Moreover, the reference population differed, so the prevalence was not comparable among studies. For case report, the prevalence calculation is inappropriate. How did authors calculated prevalence in case-control studies?

Discussion

If you want to maintain this manuscript structure, add a discussion paragraph where you discuss the limitation in comparing results produced with different diagnostic methods and highlight the potential pitfall of this review. Could be possible to aggregate data by type of study or animal species?

Conclusion

The conclusions are not supported by the results provided in the study. Do authors fill a knowledge gap with this review?

Author Response

Dear Reviewer,

Thank you for your useful comments and suggestions on the structure and content of our manuscript.

Please find a revised version of our manuscript. As requested, we modified the manuscript according to the reviewer's comments and advice.

All corrections and modifications are listed below and included in the revised manuscript.

Thank you for considering this revised version of our manuscript.

Yours sincerely,

On behalf of all authors - Dr. Marwan Osman

Reviewer's comments:

Comment 1 - The abstract focuses on the background and lack methods and results of the study. Please, consider re-writing it to give readers a summary of what’s inside the manuscript instead of an introduction to the topic.

Reply: It is worth mentioning that the current manuscript is a general review that aims to highlight the current trends in the epidemiology of latent and active BTB in the Middle East and North Africa (MENA) region. Therefore, we limited the abstract to the background and objectives of the study. However, based on the reviewer comment, we added the below sentences to add results and conclusions.

Page 1, Lines 32-38

According to PRISMA guidelines, a total of 90 studies conducted in the MENA region were found. Our findings revealed that the prevalence of BTB among humans and cattle varied significantly according to the population size and country in the MENA region. Most of the available studies were based on culture and/or PCR strategies and were published without including data on antimicrobial resistance and molecular typing. The implementation of a One Health approach is essential to prevent and control the transmission of BTB in the MENA region and beyond.

Comment 2 - Lines 39-40: “and it is the most common form of wildlife tuberculosis, with a morbidity risk reaching up to 15% of tuberculosis cases in humans” How do the two sentences relate to each other? What does wildlife TB have to do with human morbidity? Please, consider separating the sentences.

Reply: We agree with the reviewer’s comment. We moved the second part of the sentence to lines 54-55 (Page 2).

Comment 3 - Lines 65-66: “To reinvigorate and facilitate control efforts, robust surveillance programs are needed, perhaps more than ever.” Citation required or, if it is authors’ opinion, argument it in the discussion section.

Reply: We added the below reference.

Chapman, Helena J, and Bienvenido A Veras-Estévez. “Lessons Learned During the COVID-19 Pandemic to Strengthen TB Infection Control: A Rapid Review.” Global health, Science and Practice vol. 9,4 964-977. 21 Dec. 2021, doi:10.9745/GHSP-D-21-00368.

Comment 4 - Please, highlight the knowledge gap that makes your manuscript valuable for the scientific community working on this topic.

Reply: As requested, we clarified the rationale and objectives.

Page 2, Lines 80-89.

Indeed, closing epidemiologic knowledge gaps is essential for a better understanding of the risk factors for transmission of BTB among vulnerable people, to support infection prevention and control and food safety policies, and to predict future disease trends in the Middle East and North Africa (MENA) region. Therefore, the main objective of this review was to compile and discuss existing epidemiologic data on latent and active BTB in the MENA region highlight the current trends in the epidemiology of latent and active BTB in the Middle East and North Africa (MENA) region, which includes many conflict-affected and economically challenged countries; with notable deficiencies in public health and national surveillance programs for the management and control of infectious diseases, including zoonotic diseases like BTB.

Comment 5 - This section lacks details that would allow readers to reproduce the results, i.e., the complete list of research queries. Please, provide the query string used for each search engine with research terms and logic operators.

Reply: As requested, we added a flow diagram (Figure 1) describing the paper selection and inclusion/exclusion process for the review according to PRISMA guidelines.

Pages 2-3, Lines 89-104

The burden of M. bovis infections in humans and animals is not well defined in the MENA region. Therefore, we searched PubMed, Science Direct, Scopus, and Google Schol-ar databases for epidemiologic studies on BTB published between 1990 and 2021. We used a combination of words that included “Mycobacterium bovis”, “Bovine”, “MENA countries (as described previously [13])”, “Epidemiology”, “Prevalence”, and “One Health”. After importation of the search results, two authors (D. Kasir and N. Osman) in-dependently screened the citations for their relevance using the title and abstract and all qualified citations were retained for full-text assessment to confirm eligibility. Backward reference screening was performed for all articles. Data extraction was done by the same authors through a format prepared on a Microsoft Excel workbook (Figure 1). Indexed original articles in English or French, of any epidemiologic design, sampling strategy, and type (case report, longitudinal, case-control, or cross-sectional) were included. All the studies that reported original information on the prevalence of BTB in MENA countries were eligible for inclusion in the review. We excluded narrative and systematic reviews.

Comment 6 - Line 100: “unexpected cases”. Are those cases unexpected because they fell out of the M. bovis host-range? Or is it authors’ opinion? Please, argument or remove it for scientific soundness.

Reply: As requested, we replaced “unexpected” with “uncommon”.

Comment 7 - Please, provide a summary of the type of research work that were considered for this review. How many cross-sectional studies? Longitudinal ones? And case-control studies?

Reply: We added this information in tables 1 and 2 and in the main text.

Page 4, Lines 114-116

Most studies have a cross-sectional design (87%; 78/90), followed by case reports (10%; 9/90) and longitudinal studies (3%; 3/90).

Comment 8 - No risk measures are provided for risk factors. No confidence intervals are provided for any of the risk factors or for the prevalence proportions listed.

Reply: As mentioned before, this study is a general review; therefore, doing statistical analysis and assessing associations are beyond the objectives of this study.

Comment 9 - The text is structured like in a systematic review with materials and methods, results, conclusion, but it is not. Please consider restructuring the whole manuscript in a more discursive way, avoiding such structure and grouping results by means of the topic (transmission, zoonotic potential, cross-border spreading, diagnostics etc.) instead. Also, it would be useful to provide risk measures along with confidence intervals, or to suggest the degree of confidence based on the type of study from which the information was obtained.

Reply: As requested, we modified the subtitles of the different sections. This manuscript is a narrative review; therefore, doing statistical analysis and assessing associations are beyond our objectives.

Comment 10 - Lines 132-134: Please, move the references at the end of the sentence like “0.2-4.3% [41-42], 8.5-26.3% [43-44], 1.3-10.2% [45-46], 1.7-51.3% [47-48], and 0.2-20.8% [49-50]” to improve readability. Also, provide confidence intervals for each estimate, otherwise they are not comparable.

Reply: Done. As mentioned before, the objective of this review manuscript is limited to compiling and discussing existing epidemiologic data on BTB in the MENA region. We removed any prevalence on studies including one sample.

Comment 11 - Line 135: “Overall, we noticed that the prevalence has decreased in these countries over time (Table 1).” How did you come to this conclusion? Did you divide results by animal species? Did you calculate trend statistics? If so, please provide details of the method and statistical results. If not, describe clearly where you identified the decreasing trend.

Reply: It is based on our observational evaluation of studies’ prevalence results. Therefore, we have already explained that similar data are not available for every MENA country which prevents generalization of trends.

We agree with the reviewer’s comment; thus, we removed the respective paragraph.

Comment 12 - Lines 163-164: “¶Prevalence = n of M. bovis infected cases/ n of total population.” The prevalence here is difficult to understand since the population which authors refer to varied, depending on the study, from 1 to 7250, and also include non-numerical values like “dairy workers”. Moreover, the reference population differed, so the prevalence was not comparable among studies. For case report, the prevalence calculation is inappropriate. How did authors calculated prevalence in case-control studies?

Reply: We agree with the reviewer’s comment. We removed any prevalence on case report studies. Moreover, we did not find any case-control studies (It is worth mentioning that it is impossible to calculate the prevalence in such studies).

Comment 13 - If you want to maintain this manuscript structure, add a discussion paragraph where you discuss the limitation in comparing results produced with different diagnostic methods and highlight the potential pitfall of this review. Could be possible to aggregate data by type of study or animal species?

Reply: As requested before, we modified the subtitles of the different sections.

Comment 14 - The conclusions are not supported by the results provided in the study. Do authors fill a knowledge gap with this review?

Reply: Done. We modified the conclusion section.

Page 14, Lines 292-295

To our knowledge, this is the first review regarding the epidemiology of M. bovis in the MENA region. We believe that this manuscript compiles and discusses relevant data, which are of particular importance for stakeholders in the MENA region and beyond.

Reviewer 2 Report

The review explores the need for diagnosing M. bovis and its presence is compiled in the MENA region, the review states the importance of such findings in the current scenario and its implications towards One-health approach. The research question is a relevant one in current situation and has addressed it adequately. The main concept is related to zoonotic tuberculosis and in MENA region, the authors have extensively documented the prevalence of the causative agent and the methods used to diagnose them and is well compiled in the manuscript in the form of tables and references which are adequately cited and this gives the readers a clear picture of the inputs and the knowledge shared.

Author Response

Dear Reviewer,

We are pleased that you found our manuscript interesting, and we thank you for the thoughtful reading and constructive comments.

Yours sincerely,

On behalf of all authors - Dr. Marwan Osman

Reviewer 3 Report

I think that this review (which is narrative) is not correctly organized. Indeed, there is neither mention to PRISMA guidelines nor a figure showing the related flowchart. The literature search is not precisely detailed, too. If the authors' aims was a systematic review, then both the search approach and the manuscript must be completely re-organized. If so, results and discussion should be also separated.

Author Response

Dear Reviewer,

Thank you for your useful comments and suggestions on the structure and content of our manuscript.

Please find a revised version of our manuscript. As requested, we modified the manuscript according to the reviewer's comments and advice.

All corrections and modifications are listed below and included in the revised manuscript.

Thank you for considering this revised version of our manuscript.

Yours sincerely,

On behalf of all authors - Dr. Marwan Osman

Reviewer's comments:

Comment 1 - I think that this review (which is narrative) is not correctly organized. Indeed, there is neither mention to PRISMA guidelines nor a figure showing the related flowchart. The literature search is not precisely detailed, too. If the authors' aims was a systematic review, then both the search approach and the manuscript must be completely re-organized. If so, results and discussion should be also separated.

Reply: It is worth mentioning that this manuscript is a narrative review. However, we followed PRISMA guidelines to import the relevant studies.

As requested, we added a flow diagram (Figure 1) describing the paper selection and inclusion/exclusion process for the review according to PRISMA guidelines.

Moreover, we replaced the section title “Results and Discussion” with "Epidemiology of Mycobacterium bovis in the MENA region" and added new subsections with different subtitles.

Round 2

Reviewer 1 Report

Dear Authors,

Thank you for your answers. I found the manuscript's scientific soundness greatly imporved, and I think that this piece of work will help foster research in this important and topical subject. Therefore, I suggest to accept the manuscript in the present form.

Yours sincerely

Author Response

We are pleased that you found our manuscript interesting, and we thank you for the thoughtful reading and constructive comments.
Yours sincerely,
On behalf of all authors - Dr. Marwan Osman

Reviewer 3 Report

Reviewer's comments:

Comment 1 - I think that this review (which is narrative) is not correctly organized. Indeed, there is neither mention to PRISMA guidelines nor a figure showing the related flowchart. The literature search is not precisely detailed, too. If the authors' aims was a systematic review, then both the search approach and the manuscript must be completely re-organized. If so, results and discussion should be also separated.

Reply: It is worth mentioning that this manuscript is a narrative review. However, we followed PRISMA guidelines to import the relevant studies.

As requested, we added a flow diagram (Figure 1) describing the paper selection and inclusion/exclusion process for the review according to PRISMA guidelines.

Moreover, we replaced the section title “Results and Discussion” with "Epidemiology of Mycobacterium bovis in the MENA region" and added new subsections with different subtitles.

RR- The authors declare: "this manuscript is a narrative review". Therefore, it is not clear why they still keep a method section and they even included a PRISMA flowchart, which is the summary of a systematic review. 

Author Response

I tried to clarify to Reviewer 3 that although PRISMA guidelines are primarily designed for systematic reviews and meta-analyses, they can provide a useful framework for reporting the methods, results, and limitations of any type of review, including narrative reviews.  

A narrative review typically provides an overview of the existing literature on a particular topic, often with a focus on summarizing and synthesizing key findings, rather than conducting a systematic analysis of the evidence. In this sense, PRISMA guidelines can be adapted to suit the needs of a narrative review, allowing us to clearly report the methods used to identify, select, and evaluate the studies included in our review, and to present the results in a transparent and structured manner.  

I also provided a few examples of our previous narrative reviews including PRISMA.  

1) Osman, M.; Kasir, D.; Rafei, R.; Kassem, I.I.; Ismail, M.B.; El Omari, K.; Dabboussi, F.; Cazer, C.; Papon, N.; Bouchara, J.-P.; et al. Trends in the epidemiology of dermatophytosis in the Middle East and North Africa region. Int J Dermatol 2022, 61, 935-968, doi: 10.1111/ijd.15967.

2) Zakaria, A.; Osman, M.; Dabboussi, F.; Rafei, R.; Mallat, H.; Papon, N.; Bouchara, J.P.; Hamze, M. Recent trends in the epidemiology, diagnosis, treatment, and mechanisms of resistance in clinical Aspergillus species: A general review with a special focus on the Middle Eastern and North African region. DOAJ 2020, 13, 1-10, doi:10.1016/j.jiph.2019.08.007.

3) Dabbousi AA, Dabboussi F, Hamze M, Osman M, Kassem II. The Emergence and Dissemination of Multidrug Resistant Pseudomonas aeruginosa in Lebanon: Current Status and Challenges during the Economic Crisis. Antibiotics (Basel). 2022 May 19;11(5):687. doi: 10.3390/antibiotics11050687. PMID: 35625331; PMCID: PMC9137902.
